Biologging as an important tool to uncover behaviors of cryptic species: an analysis of giant armadillos (Priodontes maximus)

Cullen Joshua A. joshcullen10@gmail.com 1 2
Attias Nina 3 4
Desbiez Arnaud L.J. 3 5 6
Valle Denis 2
1 Department of Earth, Ocean, and Atmospheric Science, Florida State University , Tallahassee , FL , United States of America
2 School of Forest, Fisheries, and Geomatics Sciences, University of Florida , Gainesville , FL , United States of America
3 Instituto de Conservação de Animais Silvestres (ICAS) , Campo Grande , Mato Grosso do Sul , Brazil
4 Department of Wildlife Ecology & Conservation, University of Florida , Gainesville , FL , United States of America
5 Instituto de Pesquisas Ecológicas (IPÊ) , Nazaré Paulista , São Paulo , Brazil
6 Royal Zoological Society of Scotland , Edinburgh , United Kingdom
McElligott Alan
Electronic publication date: 2023 Jan 18
Publication date: 2023
Volume: 11
Electronic Location ID: e14726
Received 2022 Jul 14; Accepted 2022 Dec 20
Copyright: ©2023 Cullen et al.
Copyright year: 2023
Copyright holder: Cullen et al.
License: This is an open access article distributed under the terms of the Creative Commons Attribution License, which permits unrestricted use, distribution, reproduction and adaptation in any medium and for any purpose provided that it is properly attributed. For attribution, the original author(s), title, publication source (PeerJ) and either DOI or URL of the article must be cited.
License URL: https://creativecommons.org/licenses/by/4.0/

Keywords: Accelerometry, Bayesian mixture model, Habitat requirements, Neotropical savannas, Priodontes maximus, Telemetry

Funding: Houston Zoo Naples Zoo Fresno Chaffee Zoo Disney Conservation Grant Chester Zoo Jacksonville Zoo and Gardens Conservation Grants Fund RZSS La passerelle Conservation Beauval Nature Parc Animalier D’Auvergne Riverbanks Zoo & Garden Sacramento Zoo Augsburg Zoo Abilene Zoo Association Française des Parcs Zoologiques Association Francophone des Vétérinaires de Parc Zoologique Bergen County Zoological Park Idea Wild CERZA Safaris en Normandie Nature Research Wilhelma Zoologisch-Botanischer Garten Stuttgart Taiwan Forestry Bureau The Mohamed bin Zayed Species Conservation Fund Play for Nature Atlanta Zoo The Whitley Fund for Nature This study is part of the Giant Armadillo Conservation Program and has been funded by: the Houston Zoo, Naples Zoo, Fresno Chaffee Zoo, Disney Conservation Grant, Chester Zoo, Jacksonville Zoo and Gardens, Conservation Grants Fund, RZSS, La passerelle Conservation, Beauval Nature, Parc Animalier D’Auvergne, Riverbanks Zoo & Garden, Sacramento Zoo, Augsburg Zoo, Abilene Zoo, Association Française des Parcs Zoologiques, Association Francophone des Vétérinaires de Parc Zoologique, Bergen County Zoological Park, Idea Wild, CERZA Safaris en Normandie, Nature Research, Wilhelma Zoologisch-Botanischer Garten Stuttgart, Taiwan Forestry Bureau, The Mohamed bin Zayed Species Conservation Fund, Play for Nature, Atlanta Zoo and the Whitley Fund for Nature. There was no additional external funding received for this study. The funders had no role in study design, data collection and analysis, decision to publish, or preparation of the manuscript.

==============================
Advances in biologging have increased the understanding of how animals interact with their environment, especially for cryptic species. For example, giant armadillos (Priodontes maximus) are the largest extant species of armadillo but are rarely encountered due to their fossorial and nocturnal behavior. Through the analysis of speed, turning angles, and accelerometer activity counts, we estimated behavioral states, characterized activity budgets, and investigated the state-habitat associations exhibited by individuals monitored with GPS telemetry in the Brazilian Pantanal from 2019 to 2020. This methodology is proposed as a useful framework for the identification of priority habitat. Using the non-parametric Bayesian mixture model for movement (M3), we estimated four latent behavioral states that were named ‘vigilance-excavation’, ‘local search’, ‘exploratory’, and ‘transit’. These states appeared to correspond with behavior near burrows or termite mounds, foraging, ranging, and rapid movements, respectively. The first and last hours of activity presented relatively high proportions of the vigilance-excavation state, while most of the activity period was dominated by local search and exploratory states. The vigilance-excavation state occurred more frequently in regions between forest and closed savannas, whereas local search was more likely in high proportions of closed savanna. Exploratory behavior probability increased in areas with high proportions of both forest and closed savanna. Our results establish a baseline for behavioral complexity, activity budgets, and habitat associations in a relatively pristine environment that can be used for future work to investigate anthropogenic impacts on giant armadillo behavior and fitness. The integration of accelerometer and GPS-derived movement data through our mixture model has the potential to become a powerful methodological approach for the conservation of other cryptic species.

Introduction

Movement ecology provides increasingly detailed information on spatiotemporal movement patterns of species, contributing to the understanding of how animals navigate through different landscapes (Fraser et al., 2018). In particular, understanding animal behavior in preferred habitats, especially in pristine environments, can help inform effective habitat conservation measures and even policy decisions (Fraser et al., 2018). Given that animal movement patterns are affected by human activities that alter natural habitats (Tucker et al., 2018), behavioral changes from baseline activity levels in pristine environments may provide a proxy for the effect of disturbance.

Animal trajectories can be decomposed into two basic components, step lengths and turning angles. Step lengths are characterized by the distance between consecutive animal locations and turning angles are the angular deviation between headings of consecutive steps (Turchin, 1991). These components help to characterize the velocity and the tortuosity along a movement path, which are related to the activities performed over time. Additionally, biologging devices that record fine-scale movement patterns (e.g., accelerometers) can provide new insight on the spatial ecology and behavior of species. Together, these movement metrics enable the reconstruction of trajectories and the identification of short-term latent (i.e., unobserved) behavioral states that influence the fitness and distribution of individuals (Fryxell et al., 2008; Morelle et al., 2017). An individual’s behavioral state can change in response to internal and external stimuli experienced along its path (Nathan et al., 2008). Hence, exhibited behaviors may be linked not only to the species’ life history traits, but also to factors such as habitat use, foraging, predation risk and social behavior (Forester et al., 2007; Gurarie et al., 2011).

The giant armadillo Priodontes maximus Kerr, 1792 (Mammalia: Cingulata), the largest extant armadillo species, is a cryptic fossorial mammal native to South America. This species spends 80% of its time resting underground, only emerging from its deep burrows at night to feed on termites and ants (Desbiez et al., 2021). Giant armadillos select mainly forests and closed savannas to perform their aboveground activities (Desbiez et al., 2020a). In addition, these solitary mammals occur at naturally low densities (Desbiez et al., 2020b). The combination of these characteristics makes the study of giant armadillos notoriously difficult in the wild (Carter, Superina & Leslie, 2016). Importantly, this species is currently classified as “Vulnerable” (A2cd) on the IUCN Red List of Threatened Species due to habitat loss, hunting, and illegal animal trafficking (Anacleto et al., 2014).

Direct observations of giant armadillos are rare and most knowledge on their behavior comes from camera trap records and inferences based on telemetry methods (Desbiez et al., 2020c; Desbiez, Massocato & Kluyber, 2020). These studies have revealed that selection of landscape features varies over ontogeny, between males and females, and even according to activity status (activity vs rest; Desbiez et al., 2020a). However, we do not know how giant armadillos use the landscape to perform different aboveground behaviors, the diversity of their behavioral repertoire, nor the relative time allocated to each behavior.

According to optimal foraging theory (Calow & Townsend, 1981), animal behavior (including behavioral repertoire and associated activity budgets) is constrained by both energy expenditure and gain per behavior. Each behavior contributes differently to individual fitness and may be optimized in particular habitat features. Hence, discerning the behavioral processes that underlie animal movement patterns and their habitat associations may be used to investigate the effect of anthropogenic habitat degradation on individual fitness (Pagano et al., 2020; Picardi et al., 2022). In addition, the behavioral complexity and activity budget of giant armadillos are of special interest, given that this species has one of the lowest basal metabolic rates among placental mammals (McNab, 1985), which may lead to energetically constrained activity patterns and behavioral repertoire (Ancona & Loughry, 2009). By characterizing the activity budgets of giant armadillos with respect to behaviors that expend versus acquire energy, we can gain a better understanding of the ecophysiology of this species.

While there has been an increased effort to understand the movement and spatial ecology of giant armadillos within the Brazilian Pantanal (Desbiez et al., 2020a; Desbiez, Massocato & Kluyber, 2020), quantitative measures of latent behavioral states and their relationships with environmental covariates have not yet been evaluated. The estimation of latent behavioral states from animal telemetry data can elucidate which behaviors are exhibited over time and their correspondence with resources across the landscape. To better understand behavioral patterns of giant armadillos over space and time, our objectives were to (1) estimate latent behavioral states of aboveground movements from high-resolution biotelemetry data, (2) characterize daily activity budgets to understand active behavior outside the burrow, and (3) investigate the association between behavioral states and habitat types based on a combination of GPS and accelerometer data.

Materials & Methods

Study area

The study region consists of a 350 km2 area of the Brazilian Pantanal, one of the world’s largest floodplains, which has a semi-humid tropical (Aw) climate (Soriano, 2000). The landscape is a natural mosaic of different habitats, including semi-deciduous forests, closed savanna, open savanna, permanent and seasonal ponds, as well as natural and exotic grasslands. The Pantanal experiences dry (April–September) and wet seasons (October–March), which impacts the distribution of surface water across the landscape (Soriano, 2000; Harris et al., 2005). Traditional cattle ranching is practiced within the study area, although its impact is considered to be minimal because of the low density of cattle and low landscape conversion rates (Harris et al., 2005). Compared to other regions in Brazil (and the rest of South America) that are occupied by giant armadillos, this study area is considered to be relatively pristine (Venter et al., 2016). Therefore, the behavioral states and activity budget characterized by this study may provide baseline data for giant armadillos in a landscape that has low anthropogenic impact.

Capture and tagging

Seven giant armadillos were captured using funnel traps placed in front of occupied burrows, where armadillos would fall into unbaited traps upon exiting their burrow (Kluyber et al., 2020). Following capture, armadillos were anesthetized using an intramuscular injection of butorphanol (0.1 mg/kg), detomidine (0.1 mg/kg) and midazolam (0.2 mg/kg) following methods described in Desbiez, Massocato & Kluyber (2020) from May to October 2019 (Fig. 1; Table 1). Once anesthetized, armadillos were fitted with an internal VHF radio transmitter (IMP 310, Telonics, Inc., Mesa, AZ) and an external archival biologging device (TGW-4100-2, Telonics, Inc., Mesa, AZ) that included both GPS and accelerometer. The VHF transmitter weighed 38.5 g (∼1.3% armadillo body mass) and was implanted intra-abdominally while monitoring vital signs (i.e., rectal temperature, heart rate, respiratory rate, blood oxygen saturation) every 10 min, following surgical procedures in Kluyber et al. (2020). The biologging device weighed 71 g (∼2.37% armadillo body mass) and was externally attached to the animal’s carapace following methods in Silveira et al. (2009). After procedures were finished, anesthesia was reversed via an intravenous injection of naloxone (0.04 mg/kg), yohimbine (0.125 mg/kg) and flumazenil (0.025 mg/kg), and armadillos were allowed to fully recover before release (Desbiez, Massocato & Kluyber, 2020; Kluyber et al., 2020). External biologging devices were retrieved for archival data download once they fell off the armadillos, where tracking duration lasted 63 days on average (Table 1). In five individuals, biologging devices were re-deployed a second time after locating the armadillo via VHF telemetry. All animal handling was performed following the Guidelines of the American Society of Mammalogists (Sikes, 2016) and under License No. 27587 from the Chico Mendes Institute for Biodiversity Conservation, which granted permission to capture, immobilize, and manipulate armadillos.

Figure 1 Filtered GPS trajectories from each of the seven giant armadillos (Priodontes maximus) monitored in the Brazilian Pantanal between 2019 and 2020 (N = 9948 locations).

Each color represents an individual giant armadillo. The blue square on the inset map denotes the extent of the study area as shown by the base map with armadillo trajectories. Satellite imagery was accessed by an ArcGIS REST API.

Table 1 Summary data for the seven tagged giant armadillos (Priodontes maximus) from the Brazilian Pantanal (2019–2020).

Included are the ID, sex, body mass (kg), age class, number of observations after data filtering (N), date of first tracking record, date of last tracking record, and the number of days of activity monitored. The total number of filtered observations across all individuals is 9,948.

ID	Sex	Mass (kg)	Age class	N	Start date	End date	Number of active days	
Blanca	F	30.0	Adult	2,074	2019-06-23	2019-12-04	77	
Emanuel	M	34.4	Adult	1,032	2019-05-19	2019-11-02	62	
Gala	F	32.8	Adult	2,008	2019-07-20	2019-10-29	64	
Mafalda	F	27.2	Subadult	1,439	2019-05-24	2019-07-13	50	
Mazeboti	F	28.0	Adult	1,883	2019-06-23	2020-01-24	90	
Sara	F	19.0	Juvenile	596	2019-10-03	2019-11-04	32	
Tex	F	32.0	Adult	916	2019-05-23	2019-12-01	63	

For each deployment of biologging devices, GPS fixes were programmed to record every 7 min while the accelerometer was programmed to record every 5 min. The accelerometer was originally programmed for use in a separate pilot study, so the associated data have been opportunistically integrated with those from the GPS transmitter. The tri-axial accelerometer of the biologging device was programmed to summarize measurements into an “activity count” variable per time interval. This variable was calculated by detecting any values of acceleration greater than zero across each of the three axes per second of the time interval. This means that the magnitude of acceleration is not the focus of activity counts, but rather the overall frequency of movements. For example, an individual that is walking for every second of a 5 min interval (300 s) would have an activity count of 300, which would be equivalent to the same individual sprinting for the same duration. It should be noted that the accelerometer does not rely on satellite coverage and provides measurements independent of the GPS fixes.

Data preparation

We recorded in total 18,433 GPS fixes and 140,162 accelerometer measurements from all seven monitored individuals. To remove error-prone measurements that could bias the analysis and results, the dataset was filtered. First, we removed aberrant locations due to low satellite coverage (≤ 4 satellites) and, because the goal was to understand active behavior outside the burrow, we also removed activity counts recorded as zero. Then, we calculated step lengths between successive GPS locations for each individual and excluded step lengths that were larger than biologically possible for the time interval (>800 m in 7 min; 99th quantile). Moreover, GPS positions were filtered to only include observations successfully sampled at 7 ± 1 min time intervals. Since activity counts were recorded at a different time interval than GPS locations (5 vs 7 min, respectively), only activity counts recorded within 1 min of GPS positions were retained to maintain comparable measurements. Step lengths were converted to speed by dividing its value by the time interval between consecutive locations (e.g., 7 min). After completing these filtering steps, a total of 9,948 observations of both location and activity count were retained for further analysis (Table 1; Fig. 1).

Latent behavioral state estimation

We relied on activity count, speed (as calculated above), and turning angle for the estimation of latent behavioral states using the non-parametric Bayesian mixture model for movement (M3; Valle et al., 2022). Each variable was discretized into bins as required by the model, where the number and width of bins were selected to characterize the continuous distributions in as few bins as necessary (Cullen et al., 2022). Activity count ranged from 1 to 300 and was discretized into six bins of equal width. Speed ranged from 0 to 2.13 m/s and was discretized into 7 bins. Since 99% of speed observations were recorded below 1 m/s, 6 bins of equal width were used to discretize speed from 0 to 1 m/s. All observations greater than 1 m/s (up to 2.13 m/s) were assigned to the 7th bin. Turning angle ranged from −π to π radians and was discretized into 10 bins of equal width (Fig. 2).

Figure 2 Density plots of each of the three movement metrics analyzed by the Bayesian mixture model for movement (M3), pooled across all seven monitored giant armadillos (Priodontes maximus).

These density plots show the original distribution of values for activity counts, speed, and turning angle as derived from the biologging devices. Values used to bin each of the movement metrics into discretized variables (as required by M3) are denoted by vertical dashed lines.

The three discretized variables (i.e., activity count, speed, and turning angle) were analyzed using M3 via the package ‘bayesmove’ v0.2.0 (Cullen et al., 2022) in the R statistical software (v4.0.2; R Core Team, 2020). This model clusters observations (pooled across all individual tracks) into an unknown number of discrete latent behavioral states. Using a penalizing Bayesian prior, we only need to specify the maximum number of possible behavioral states and the model estimates the most likely number of states assigning these to observations. The model was run using 20,000 Markov chain Monte Carlo (MCMC) iterations where the first 10,000 iterations were treated as the burn-in, the maximum number of possible states was set to 10, and the hyperparameter α was set to 0.1. Convergence to the posterior distribution was assessed by evaluating the log likelihood trace plot. This model took 18 min to run on a 2.6 GHz i7 CPU with 16 GB RAM.

The number of latent states was determined by assessing the probability of state assignments as well as the biological relevance of the state-dependent distributions. Specifically, the most likely number of states was estimated by calculating the average probability that observations belonged to a particular state and selecting the set of states that together represented ≥90% of all observations. Following recommendations by Pohle et al. (2017), state-dependent distributions of activity counts, speed and turning angles were inspected so that only biologically interpretable states were retained. To ensure high certainty of behavioral state assignments, ≥75% of all posterior estimates of a given observation needed to belong to a single state. If none of the states for a given observation fell above this threshold, the observation was labeled as ‘unclassified’. The impact of ancillary activity count measurements on estimating the number of likely states was evaluated by running the M3 model again using the same parameterization, but only analyzing the speed and turning angle data streams (i.e., activity counts from the accelerometer were discarded).

Activity budgets

To determine diel patterns of behavior, activity budgets were calculated with all individuals pooled together. Hourly counts of biologging transmissions were calculated, as well as the relative proportion of estimated behavioral states exhibited for each hour. Unclassified observations were omitted from the quantification of activity budgets to emphasize the relationships among classified states. Diel activity patterns were qualitatively examined to determine when giant armadillos were most active and to explore temporal patterns in the estimated behavioral states.

Behavioral state-habitat associations

To understand the relationship between the estimated behavioral states and the habitat used by giant armadillos within the study area, we extracted seasonal time-matched land use/land cover (LU/LC) classes at each location of each individual. The LU/LC data was obtained via supervised classification of Landsat 8 imagery from 2018 as described in Desbiez et al. (2020a). Four classes of LU/LC were identified separately for the dry (April–September) and wet (October–March) seasons: forest, closed savanna, open savanna, and floodable grasslands (Fig. 3). Floodable grasslands are completely or partially flooded during the wet season and covered by grassland and sparse shrubs during the dry season. LU/LC was extracted using a 30 m buffer around each GPS position using the R package ‘raster’ v3.4-10 (Hijmans, 2021). This buffer was adopted to capture local habitat heterogeneity for a given location and to account for GPS location error. Since LU/LC is a discrete variable, the proportion of each class within the buffer was calculated for all observations.

Figure 3 Landscape mosaic of land use/land cover (LU/LC) for the dry (April–September) and wet (October–March) seasons of the study region in the Brazilian Pantanal.

To assess the behavioral state-habitat associations across the local landscape, we fitted a mixed effect Bayesian multinomial logistic regression to evaluate how the prevalence of each state was influenced by LU/LC (Wilkinson et al., 2019). This model estimated the probability of exhibiting each of the behavioral states with respect to a reference state while accounting for differences among individuals. Fixed effects were included for the proportion of the forest, open savanna, and floodable grassland classes, whereas closed savanna was chosen as the baseline LU/LC class and therefore was omitted from the list of covariates. Quadratic terms were included for each LU/LC class to account for non-linear relationships. Armadillo ID was treated as a random effect on the intercept to account for individual heterogeneity. The slowest state estimated by the model (labeled ‘vigilance-excavation’) was treated as the reference state; additional details about the estimated states can be found in the Results. To fit this model, we used the R package ‘brms’ v2.16.3 (Bürkner, 2017) with a categorical distribution and logit link function, where the no-U-turn sampler (NUTS; Hoffman & Gelman, 2014) from Hamiltonian Monte Carlo was used via Stan (Stan Development Team, 2020) to obtain posterior distribution samples. Regularized priors were used for model parameters to keep estimates within the plausible domain space, which was verified by a prior predictive check (Wesner & Pomeranz, 2021). We ran 4 chains with a warmup of 1,000 iterations and retained 1,000 iterations from the posterior distribution, resulting in a total of 4,000 samples from the posterior. Model convergence was assessed by inspecting trace plots, effective sample size, and R ˆ (Vehtari et al., 2021) for each parameter, while model fit was assessed through a posterior predictive check (Gabry et al., 2019; Gelman et al., 2020). After running the model, trace plots appeared to exhibit stationarity, the effective sample size was near the total number of posterior samples, and R ˆ was equal to 1.00 for all estimated parameters, indicating model convergence.

Results

Latent behavioral state estimation

The mixture model suggested that four states were likely present for these data (accounting for 98% of all observations) based on the aforementioned threshold and inspection of the state-dependent distributions. Based on prior direct observations of giant armadillos, we decided to label these states as ‘vigilance-excavation’ (VE), ‘local search’, ‘exploratory’, and ‘transit’ (Fig. 4). The VE state is characterized by relatively low activity counts, low speed, and high turning angles (near −π or π radians), which is congruent with the movements observed when armadillos are exhibiting vigilant behavior or when excavating either termite mounds or new burrows. The local search state is nearly identical to the VE state for distributions of activity counts and speed, but has a more uniform distribution of turning angles that is expected to occur when armadillos are performing an area-restricted search for food. The exploratory state is characterized by higher activity counts, greater speed, and lower turning angles (near 0 radians) than either the VE or local search states, which reflects a relatively faster and more directed movement that can be associated with ranging movements, most likely to find food. Lastly, the transit state exhibited the greatest activity counts, greatest speeds, and lowest turning angles, reflecting fast and directed movements with the primary purpose of displacement between two areas, as movement speed and high directionality would unlikely allow foraging or other activities during this behavioral state.

Figure 4 State-dependent distributions of activity counts, speed, and turning angle estimated by the mixture model for each of the four estimated behavioral states (vigilance-excavation (VE), local search, exploratory, and transit).

Continuous values (ranging from minimum to maximum values) of activity counts, speed (m/s) and turning angle (radians) were discretized into bins. Activity counts were provided by an accelerometer, while speed and turning angle were derived from individual GPS trajectories from seven giant armadillos (Priodontes maximus) monitored in the Brazilian Pantanal between 2019 and 2020.

Of the 9,948 observations analyzed by this model, 38% were labeled as ‘unclassified’ since the state assignments from the posterior distribution did not overwhelmingly belong to a single state. The remaining 6,180 observations were classified as one of the four estimated behavioral states.

Activity budgets

Giant armadillos were most active during the middle of the night and displayed notable differences in the temporal patterns exhibited by each behavioral state. The greatest number of joint GPS and accelerometer measurements from the biologging device (>600 measurements per hour) occurred between 20:00 and 01:00 across all tagged individuals (Fig. 5A). However, it appears that armadillos can be active at any time throughout the course of the night, including dusk and dawn. On average, armadillos dedicated most of their active time to local search (44%) and exploratory states (33%), whereas VE (19%) and transit states (4%) were observed less frequently. When evaluated on an hourly basis, the local search and exploratory states showed minimal changes in their frequency throughout the night (Fig. 5B). By comparison, the VE state was exhibited in greatest proportions at the very first and last hours of recorded activity, whereas the transit state was exhibited in greatest proportion during the second to last hour of activity (Fig. 5B).

Figure 5 Activity budget of giant armadillos (Priodontes maximus; N = 7) along the diel cycle.

(A) General activity budget per hour of the day characterized by the total number of activity observations pooled across all monitored individuals. (B) Temporal patterns of behavior-specific activity characterized by the proportion of each behavioral state observed at each hour of the day. Grey shaded areas represent the maximum period from dusk to dawn (i.e., night) for the study region in the Brazilian Pantanal during the study period (2019–2020). The vigilance-excavation state is abbreviated as ‘VE’.

Behavioral state-habitat associations

Strong relationships were found between specific land cover classes and behavioral states. The likelihood of displaying the VE state increased when armadillos were in areas with high proportions of open savanna and floodable grasslands (>80% cover) or along forest edges with closed savannas (i.e., intermediate proportions of forest; Fig. 6, Table S1). The probability of performing the exploratory state increased in areas with high proportion of forest cover and low proportion of closed savanna, and in areas with low proportion of forest cover and high proportion of closed savanna, with lower probability in areas with intermediate coverage (i.e., 25–75%) of both LU/LC classes (Fig. 5). Additionally, armadillos displayed a low overall probability of being in a transit state and no significant trends associated with this behavioral state were detected. Finally, although local search behavior was the most likely state to be exhibited across each of the evaluated LU/LC classes, a negative relationship was found with increasing proportions of forest, open savanna, and floodable grasslands (Fig. 6, Table S1). These results imply that local search was more likely in areas with high proportions of closed savanna. Posterior distributions for each of the parameter estimates from the multinomial regression are shown in Fig. S2 and the probabilities of exhibiting each state when in 100% of each LU/LC class are shown in Fig. S3.

Figure 6 Probability of giant armadillos (Priodontes maximus) exhibiting each behavioral state as a function of the proportion of land-use/land cover (LU/LC) classes in the area used.

Proportions of LU/LC classes are defined in relation to Closed Savannas (e.g., when animals are in areas with 25% of the plotted LU/LC class, the remaining 75% of the area is covered by Closed Savanna). Behavioral states are classified as: Vigilance-Excavation (VE), Local Search, Exploratory and Transit. Plot curves represent the marginal effects of the Bayesian multinomial logistic regression. Solid lines display the average response, and shaded regions denote 50% (dark) and 95% (light) credible intervals.

Discussion

Identification of behavioral states in a cryptic species

Four behavioral states appeared most likely from the analysis of activity counts, speed, and turning angles in fine-scale movements. Each of the identified states appeared to reflect behaviors of giant armadillos previously observed in the field (pers. obs.), supporting the use of these results to interpret the behavior of this cryptic species. However, we could not independently verify the true behaviors exhibited for each of the states estimated by the model. Therefore, future work (for example using camera traps or the study of captive animals) is needed to validate these estimated behavioral states. Additionally, we acknowledge the potential for negative health outcomes when performing invasive procedures on wildlife, however, we have not documented any such negative effects on animal health in the past 12 years of ongoing work on this species. Future studies on giant armadillos would likely remain successful when attaching only external tags.

Our results also highlight the contribution of ancillary movement measures, such as accelerometry activity counts, for the identification of latent behavioral states. When analyzing only speed and turning angles, the non-parametric Bayesian M3 method was only able to identify three behavioral states, where ‘exploratory’ and ‘transit’ behaviors appeared to be grouped together (Fig. S1). These results suggest that the activity counts accelerometer data was critical for the successful discrimination of the “exploratory” and “transit” behaviors. Distinguishing these two behaviors is especially important in studies such as ours that aim to relate behavioral budgets with energetic constraints, as the former behavior is related with energy acquisition while the latter relates to direct displacement from one point to another and energy expenditure. Even though the activity counts measure from the accelerometer provides less detailed information when compared to tri-axial accelerometry, its inclusion as an additional data stream (beyond speed and turning angle) appears to have assisted in greater discrimination among behavioral states. We believe that, had our accelerometers provided more detailed information instead of a summary metric (i.e., number of seconds with activity), we would have been able to reduce the proportion of unclassified records or even been able to distinguish additional behaviors. More research is needed to determine if this is likely to be true.

Activity budgets

This study reveals that giant armadillos are most active during the middle of the night (20:00–01:00; Fig. 5A). Although we detected armadillo activity at varying levels throughout the entire night, recorded activity only lasted approximately 5 h on average each day, corroborating the results found by Desbiez et al. (2021) using a different set of data and analytical approaches in the same study region.

Proportions of local search and exploratory states were both high throughout the entire nightly activity period and together represented 77% of all the classified observations. Based on field observations, these two behaviors may be associated with foraging activities. Previous studies have shown that armadillo activity budgets are mostly oriented towards energy acquisition, with very limited social interactions (Ancona & Loughry, 2009; Ancona & Loughry, 2010). Indeed, the only other mammals that dedicate such high proportions of their activity budgets to foraging activities are the nine-banded armadillo Dasypus novemcinctus (77–100%), the burrowing ground squirrel Urocitellus townsendii (65–97%), and species of arboreal marsupials from the genus Petaurus (66–85%; Sharpe & Van Horne, 1998; Jackson & Johnson, 2002; Ancona & Loughry, 2009). Giant armadillos are large mammals (32–60 kg), with short activity periods, low basal metabolic rates and an energy restricted diet composed mostly by ants, termites and other invertebrates, which are scattered across the landscape (McNab, 1985; Ancona & Loughry, 2009; Desbiez et al., 2021). Hence, it is not surprising that most of its activity budget is spent on search and acquisition of food (i.e., local search and exploratory states).

The relatively high proportions of the VE state during the very first and last hours of the activity period are likely associated with vigilance behavior upon leaving the burrow and excavating new burrows at the end of the night (Fig. 5B). Low predation pressure on giant armadillos may be associated with the relatively small proportion of time spent on vigilance throughout the night, especially when away from its shelter (Aya-Cuero, Rodríguez-Bolaños & Superina, 2017; Pasa et al., 2022). Hence, the consistently low levels of the VE state observed throughout the night are likely reflective of termite mound excavation while foraging, which requires less persistent digging but may generate similar movement patterns to those observed during vigilance or the digging of burrows.

Individuals rely on spatial memory or cognitive maps to survive (sensu Gallistel, 1994; but see Bennett, 1996). Relatively rapid and directed movement is observed when cognitive mechanisms (e.g., memory) are employed to locate resources (Polansky, Kilian & Wittemyer, 2015). The relatively higher proportion of transit behavior near dawn could be associated with individuals quickly returning to burrows, a key sheltering resource for this species (Fig. 5B). Giant armadillos are range residents and use the same area for many years (Desbiez, Massocato & Kluyber, 2020; Desbiez et al., 2021). Importantly, they are known to reuse burrows for two to three consecutive nights and sometimes also after long periods (pers. obs.). Hence, this behavior could indicate that these important sheltering structures might be somehow mapped by an individual within its home range.

State-habitat associations

We found associations between certain latent behavioral states and particular habitats, which provides new insight into the species’ movement ecology and habitat requirements. Specifically, we determined that the behavioral states (estimated via the characterization of acceleration, speed, and turning angle) of giant armadillos varied with respect to external stimuli experienced along its trajectory. Giant armadillos have been documented to excavate burrows and rest in habitats with denser forest vegetation above the floodplain, while typically selecting closed savanna habitat during activity (Desbiez et al., 2020a). These patterns of habitat selection are supported and explored in further detail by the state-habitat relationships characterized here.

The likelihood of displaying the VE state increased when armadillos were in areas with high proportions of open savanna and floodable grasslands (Fig. 6). Like other species, the use of dense-cover habitat by giant armadillos has been associated with reduced predation risk (Desbiez et al., 2020a). Hence, when reaching more open areas with higher perceived risk, animals would tend to increase vigilance behavior. Nevertheless, the likelihood of presenting a VE state was also higher in transition areas between forest and closed savanna (Fig. 6), which may reflect an increase in foraging-related excavation behavior in search for ground-dwelling invertebrates. The giant armadillo’s foraging behavior could also justify the high probability of the local search state when individuals are surrounded predominantly by closed savanna (Fig. 6). This species forages preferentially in closed savanna habitats, which are always adjacent to forests in the Pantanal (Desbiez et al., 2020a). Furthermore, the association of local search and VE states to closed savanna and forest edges, respectively, is also corroborated by recent dietary studies based on stable isotope analysis, which show that most giant armadillos feed on invertebrates from both open and closed habitats (i.e., forests and savannas; Magioli et al., 2022). Finally, exploratory behavior was most likely to be present when individuals were surrounded by high proportions of forest or closed savanna (Fig. 6). Hence, the exploratory behavior is likely part of the search process for resources, such as food and shelter, that can be found mainly in forests and closed savannas (Desbiez et al., 2020a). Since the exploratory and local search states were both relatively similar in their state-dependent distributions, activity budgets, and associations with land cover classes, future work that uses original data (i.e., not summarized) from tri-axial accelerometers may improve the classification of these two states.

Giant armadillos have also been shown to use floodable grasslands during the dry season, which was presumed to increase their mobility due to a lack of obstacles such as dense vegetation (Desbiez et al., 2020a). In our analyses, armadillos presented a low overall probability of displaying the transit state and no significant relationships with LU/LC classes were detected. However, there was relatively high uncertainty around these estimates, likely resulting from the few observations classified as transit by the model (n = 213; 3.4%). It appeared that higher speed transit behavior was more likely to be exhibited in high proportions of floodable grasslands compared to other habitats, corroborating the previous hypothesis that this food-poor habitat may be selected to increase an individual’s mobility. Nevertheless, this pattern should be further explored.

Implications for conservation planning

The identification of behavioral patterns generated by the analysis of multiple individuals allows us to bridge movement ecology with population processes (Morales et al., 2010) that are fundamental for conservation planning. This study links latent behavioral states to occupied habitat, providing critical insight on giant armadillo interactions with the surrounding landscape. These findings are complementary to previous studies on resource selection in giant armadillos within the Brazilian Pantanal and neighboring biomes (Desbiez et al., 2020a; Ferraz et al., 2021; Magioli et al., 2022), while providing novel insights on the types and proportions of behaviors exhibited in a pristine region.

The estimated behavioral state-habitat associations can be used to inform conservation planning and decision-making as part of current conservation efforts. Our results provide baseline information for how the species moves within a relatively pristine environment that consists of a naturally fragmented landscape with low human pressure. Giant armadillos search for resources and feed primarily in savannas and forests, while sheltering in areas of forest. Furthermore, juveniles (up to 7−8.5 years old) rely almost exclusively on forest, making this habitat key for population stability and growth (Desbiez et al., 2020a). Nevertheless, the grassland matrix is used for transit and does not present biophysical resistance to individual dispersal movement.

Giant armadillos tolerate some degree of habitat disturbance (Silveira et al., 2009). Nevertheless, in naturally fragmented floodable savannas, each individual requires large areas (25 km2) that minimally overlap with that of conspecifics (Desbiez et al., 2020b). Future work on giant armadillos tracked in regions impacted by natural and anthropogenic disturbances (e.g., agriculture, urban development, deforestation, wildfire) can then be assessed in the context of the aforementioned behavioral state-habitat associations. For example, in the neighboring Cerrado savanna biome, only 69 remnants of suitable habitat larger than 25 km2 persist and are surrounded by agricultural matrix (Ferraz et al., 2021). The methodological approaches adopted here could be used to understand how giant armadillos can persist in fragments within a human modified matrix with moderate to intense management practices. Due to its large area requirements and the accelerating rate of land-use change throughout its range, giant armadillos need conservation measures that go beyond the creation of protected areas (Desbiez et al., 2020a). Hence, conservation efforts that increase connectivity among remnants of habitat will be key for species recovery and persistence in this biome and throughout its range.

Changes in behavioral repertoire, activity budget, and habitat associations may help conservationists to better understand the consequences of anthropogenic impacts on individual behavior and fitness. For example, we would expect an increase in the probability of exhibiting a vigilance state in areas of higher perceived risk (e.g., areas close to highways, with hunting activities, or with domestic dogs). These perceived risks could also affect the species’ activity budget in areas with higher human disturbance. In addition, this multi-faceted approach could be used to better understand the degree of impact from different types of anthropogenic land-use. For example, state-habitat associations can be investigated to determine whether a land-use class may be categorized as a resource (i.e., high probability of resource acquisition behaviors), corridor (i.e., high probability of dispersal behavior), or a risk (i.e., mostly avoided by individuals). Furthermore, this approach may serve as an extraordinary tool to evaluate the effectiveness of conservation actions, such as the definition of ecological corridors and priority areas.

Conclusions

Our results suggest that giant armadillos primarily exhibit behaviors related to foraging throughout the course of their active nocturnal periods, which are particularly associated with closed savanna habitat. These findings provide baseline information for future investigations that explore giant armadillo movements in human-impacted landscapes. While, the success of conservation measures is usually quantified through population growth via survival, resource acquisition, and reproductive success (Lacy, 2019), these metrics can be demanding to obtain and are especially difficult to gather for cryptic, long-lived species with low reproductive rates like giant armadillos. Hence, the identification of latent behavioral states, their associations with landscape features, and activity budgets can provide a short-term, powerful, and feasible indicator of success for conservation actions. We believe that the methodological approach presented here provides researchers and conservationists with a useful tool to plan and evaluate management actions towards the conservation of cryptic species.

Supplemental Information

Supplemental Information 1 Supplemental Figures and Tables

Click here for additional data file.

Supplemental Information 2 ARRIVE 2.0 Checklist

Click here for additional data file.

We would like to thank G. Massocato and D. Kluyber for their essential contribution to this work by performing animal capture, tagging, and monitoring in the field. We are grateful to the Fazenda Baía das Pedras owners (R. and C. Jurgielewicz) for their hospitality, generous support, and permission to work on their land.

Additional Information and Declarations

Competing Interests

Author Contributions

Animal Ethics

Field Study Permissions

Data Availability

The authors declare there are no competing interests.

Joshua A. Cullen conceived and designed the experiments, analyzed the data, prepared figures and/or tables, authored or reviewed drafts of the article, and approved the final draft.

Nina Attias conceived and designed the experiments, performed the experiments, authored or reviewed drafts of the article, and approved the final draft.

Arnaud L.J. Desbiez conceived and designed the experiments, performed the experiments, authored or reviewed drafts of the article, and approved the final draft.

Denis Valle conceived and designed the experiments, analyzed the data, authored or reviewed drafts of the article, and approved the final draft.

The following information was supplied relating to ethical approvals (i.e., approving body and any reference numbers):

Chico Mendes Institute for Biodiversity Conservation issued a permit to conduct animal handling and tagging (permit number 27587).

The following information was supplied relating to field study approvals (i.e., approving body and any reference numbers):

A field permit was not required to conduct research at our field site since these were privately owned lands. We therefore obtained permission from the landowners (Rita and Carlos Jurgielewicz), whom we mention in the Acknowledgements.

The following information was supplied regarding data availability:

All code used to process and analyze the giant armadillo data are available on Zenodo: Josh Cullen. (2022). joshcullen/acceleration: Third release of R scripts for analyses (v2.1.0). Zenodo. https://doi.org/10.5281/zenodo.7278215.

The giant armadillo GPS telemetry data is available at Movebank: https://www.movebank.org/cms/webapp?gwt_fragment=page=studies,path=study2402574829.

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
