# Peer review of "Biologging as an important tool to uncover behaviors of cryptic species: an analysis of giant armadillos (Priodontes maximus)"

_PeerJ, doi:10.7717/peerj.14726_

## Round 0.1 · original submission · Major Revisions

Thank you for submitting this interesting study to PeerJ. I regret that I am unable to accept the manuscript for publication, at least in its present form. However, I am prepared to consider a new version that carefully takes into account the suggested revisions. The reviewers liked many aspects of your study but also highlighted some concerns. These need to be addressed in detail in the new version. Such a revised manuscript may be reviewed again and there is no guarantee of acceptance. When you revise your manuscript, please prepare a detailed explanation of how you have dealt with all of the reviewers' and my own comments.

Please provide more exact details about the ethics of the research, and the welfare of the animals involved. See: https://peerj.com/about/policies-and-procedures/#animal-research. As readers, we cannot really be expected to read other papers for vital details about the research presented in this study (e.g. lines 132-136). Also: "We require all authors to comply with the 'Animal Research: Reporting In Vivo Experiments' ARRIVE 2.0 guidelines"

·

Basic reporting

The manuscript is overall very well organized and written. The English language is really good.

Although I agree that your methodology could be used on other species in the future, your current title makes it sound like you worked on a range of cryptic species whereas you studied one specific species.

The Abstract provides all the necessary information to understand the manuscript in a few sentences.

The Introduction provides sufficient background/context and states clearly how the manuscript will contribute to the field.

L229. It is the first time you mention VE in your manuscript. At this point, readers do not know what you are talking about and why it is the reference state. I advise the authors to make the necessary changes.

L381. Explain C3 and C4 resources.

Overall, the figures and tables look good, as do their descriptions.
Figure 2. Could be worth considering giving details about the seasons again: dry (April – September) and wet (October – March).
Table 1. I personally would put N further right so that all the information about the individuals is together (ID, sex, age) and then the information about the collected data.

The article provides relevant literature. I advise the authors to stick to one way of citing their references in their manuscript (i.e., choosing between “and” or “&”).
L80-83. Do you have a reference?
L333-337. You give information on several species but only cite one study on nine-banded armadillos?
L340. The citation needs to be modified to be in chronological order.
Could you please add the corresponding c and d in your references after Desbiez et al. 2020? (L508 and 512)

Although it could be because I have never used this website before, I could not find the GPS telemetry data on Movebank. The study details are clearly given, but I cannot view or download any data. Data on Zenodo is easily downloadable. I thank you for providing the supplemental files. The ARRIVE checklist is consistent with the manuscript.

Experimental design

Mat&Met. Because we cannot expect readers to read all the cited articles, I advise the authors to include the essential details of the cited references in their manuscript (e.g., capture and anesthesia “following methods described in Desbiez et al. (2020c)” or implantation of VHF transmitter “following surgical procedures in Kluyber et al. (2020)”).

Would it be possible to add picture(s) of the animal wearing the devices?

Although I thank you for providing the Licence number and the actual document that allowed you to perform your study, I do not know Portuguese, therefore I am unable to assess the ethical standards.

Could you add an explanation as to why you sometimes have squared variables? (Fig S1 and Table S1).

Validity of the findings

I found the topic of this manuscript very interesting. The study of wild individuals is most often challenging, even more so with cryptic species. Technological advances open doors to novel methodologies that can allow us to learn more about animals’ behavior which can then have important implications for future conservation efforts.

The conclusion is clear and meaningful. The proposed future directions appear relevant.

Additional comments

I thank the authors for their work. The manuscript is clearly written and enjoyable to read. I mostly noticed minor errors which should be verified before acceptance.

Reviewer 2 ·

Basic reporting

This is a clear manuscript reporting behavioural data on giant armadillos by using biologging. The references need to follow the journals style, i.e. use & instead of and when citing in text 2 or 3 authors. Use a comma to separate authors last name from year, i.e. Line 60 (Turchin, 1991) and also add a comma after et al.
More importantly there are two references that need clarification. Are these Desbiez et al., 2020c and Desbiez et al., 2020d?:

Desbiez ALJ, Massocato Gabriel F., Attias N, Cove M V. 2020. Comparing density estimates from a short-term camera trap survey with a long-term telemetry study of giant armadillos (Priodontes maximus). Mastozoologia Neotropical 27:241.246. doi:10.31687/saremMN.20.27.2.0.08.

Desbiez ALJ, Massocato Gabriel Favero, Kluyber D. 2020. Insights into giant armadillo (Priodontes maximus Kerr, 1792) reproduction. Mammalia 84:283.293. doi:10.1515/mammalia-2019-0018. (I assume this paper is Desbiez et al 2020c since the capture and anaesthesia protocol is provided).

The figures are relevant and informative
The research questions are clearly defined, there is no hypothesis.

Experimental design

Material and methods:
The study has an authorization for the work with animals provided by the Chico Mendes institute for biodiversity conservation (N°27587). This license is included in the material provided by the authors and includes permits for capture and the use of internal monitoring systems.
L128. Please check the reference list since Desbiez et al. 2020c is not listed and is required to check the capture system and the anaesthesia protocol. Otherwise explain i brief the capture-handling and drug protocols used as the surgery.

L142-145. The authors explain that the GPS were programmed to record every 7min while the accelerometers every 5min. Could you please explain why they were not programmed to record during the same periods of time? Considering that later on it is explained that data had to be filtered to include successful samples = 7 ± 1 min time intervals (L163). Also for how many days in average was each giant armadillo followed. This information appears in Table 1, but should be explained in the methodology.

In line 128 it is explained that captures were done between May and October of 2019, would this mean that data from some Armadillos are from different seasons? How can this affect the behaviour of the animals? In Table 1 the start and end dates are provided, but this should be discussed.

In the results section the authors report 4 states according to their mixture model, but then in line 309 they report 3, with “exploratory” and “transit” behaviours grouped together, this re-grouping of the behaviours should be better explained in the results section, in particular considering the importance of being able to discriminate both behaviours as explained by the authors.

L270: For the activity budget it would be nice to first have a brief description of the total time budget of the armadillos divided in active and inactive (in the burrows) time. This considering that the diel cycle is presented in the figures and also in the methodology it is explained that diel patterns will be calculated. Also, this would allow to better understand the percentage of time dedicated to the behaviours described during the active time, were these percentages calculated over 5hours of activity? (this information is provided in the discussion section, but should appear earlier in the results section) Or over 12h? as shown in figure 4. Again, in this section the percentage of time dedicated to 4 behaviours are provided and not 3.

L304-305. The authors report that the 4 behaviours identified appear to reflect the field observations from the author, I think it would be appropriate to include a limitations of the study section in order to clarify that a validation of this system needs to be done.

Validity of the findings

The study provides novel information on the use of biologgers to estimate some behavioural states in giant armadillo. Still it should be considered the invasiveness of the method and how this reduces the practical use of this methodology and the information retrieved. In particular when working with endangered species this invasive procedures should be avoided since they could cause negative health or behavioural implications and even death.I suggest including some limitations in terms of welfare implications.
Some limitations of the methodology should be included in the discussion section. This methodology still needs validation to understand if the model used is accurate, here the authors explain that it does resemble their field observations but a validation method is required.
The association between the behavioural states and habitat is probably the most interesting result in terms of conservation strategies and it would be interesting in the future to understand how human disturbance changes the behaviours of these animals, although this was not the aim of the present study.

·

Basic reporting

Overall, the manuscript is well written, and its findings will advance the knowledge about the spatial ecology and behavioral ecology of giant armadillos within the Brazilian Pantanal which are also fundamental for conservation planning.

Experimental design

No comment.

Validity of the findings

No comment.

Additional comments

Line 69: Can you please verify if this factor is relevant to be mentioned here? That one caught my attention because you will inform later that armadillos are 'solitary mammals' in the line 75.
 Lines 87—89: Was that lack of information solved after concluding this telemetry study? Otherwise, I believe it would be relevant to link those pieces of information to the goals of this study as listed in the lines 106 to 109.
 Line 95: Is this the primary reason why you conducted this research on armadillos? As you repeated this piece of information twice in this Section (also, line 54), it adds an opposite idea to what is said in these lines: Line 122: "this study area is considered to be relatively pristine"; Line 124: "...that has low anthropogenic impact". Otherwise, please consider clarifying the primary reason why you conducted this study.
 Line 97: Why these pieces of information are placed here? The importance of these facts is not so clearly linked to the ideas of the previous sentences.
 Line 98: Please check a comment made in the line 312.
 Line 107: Please consider adding this piece of information that I retrieved from the results section for the sake of clarity: (line 157) "to understand active behavior outside the burrow".
 Line 111: I cannot find a list of the armadillos' behavioral states in the 'Materials and Methods' section. As it was mentioned four behavioral states in the lines 81 to 83 in the 'Introduction' section, I believe you are referring to those ones. So, can you please insert that list of behavioral states in this section for the sake of clarity?
 Line 128: Has this study occurred only during the dry season?
 Line 131: Were adults those seven armadillos? If so, what is an average body weight of an adult armadillo?
 Line 137: Why those five individuals were able to get rid of the biologging devices? I believe it is a piece of important information to infer about the overall state of the armadillos' welfare.
 Line 157: Why is this goal not clearly stated in the introduction?
 Line 161: Please consider adding comments about this limitation in the discussion section.
 Line 171: In the lines 81 to 83, it is said: "directional displacement, vigilance behavior while standing on their hind feet, moving relatively slowly using their sense of smell to assess the environment, digging termite mounds and portions of soil for feeding, or digging burrows for sheltering". If those armadillo's behavioral states are already stated in the introduction, why were those four behaviors not added here? I would expect that those same four behaviors would be mentioned here.
 Line 180: I believe that you should call figure 3 (or maybe a new one) here. The reason for that is because it took me a while to find out what figure portrayed those values informed from lines 176 to 180. I also believe that figure 3 helped me to figure out all those pieces of information that were placed here.
 Line 181: Please consider adding some comments about this limitation in the discussion section.
 Line 193: Why is this information relevant?
 Line 208: What are those behavioral states implied here?
 Line 229: Please consider defining the 'VE state' here.
 Line 252: Are those states part of the armadillo's behavioral repertoire? In lines 81 to 83, why do you state that armadillo's behavioral states are "directional displacement, vigilance behavior while standing on their hind feet, moving relatively slowly using their sense of smell to assess the environment, digging termite mounds and portions of soil for feeding, or digging burrows for sheltering"? What about those other terms: ‘vigilance-excavation’ (VE), ‘local search’, ‘exploratory’, and ‘transit’? Where do those terms come from?
 Line 274: The letter 'a' is missing in fig. 4.
 Line 284: In this whole paragraph, those results are not well linked to the figures and tables cited here. Can you please check it out?
 Line 287: Where are shown such ‘areas of transition’ in fig. 5?
 Line 288: There are some values shown in the 'intercept' rows in the tab. S1. However, the 'VE' state is not included in the tab. S1. However, the 'VE' state is not included in the tab. S1. It means that you cannot call it here. Can you please check it out?
 Line 291: I still do not understand what the intercept area is. For this reason, I just wanted to make sure I understood this tab. S1 well. So, can you confirm if such "intermediate coverage" corresponds to the row in the tab. S1 where the probability of an armadillo performing the exploratory behavior in an "intercept area" is 0.78?
 Line 301: Is it correct to subdivide this section?
 Line 312: This goal is not put clearly in the introduction. Can you please add a sentence to inform this goal clearly in the introduction (lines: 97—99)?
 Line 319: It is missing the word “to”.
 Line 325: The letter 'a' is missing in fig. 4.
 Line 330: I only now found out that the armadillo's behavioral repertoire is meant to be inferred from each of the four behavioral states, which are vigilance-excavation, local search, exploratory, and transit.
 Lines 333, 337 and 340: Please verify which one is correct.
 Line 345: Please consider citing any work to support this statement.
 Line 349: Is this the reason why 'VE' state was not separated into two categories?
 Line 361: I cannot find any additional comment concerning these statements made in the introduction: Line 66: "an individual’s behavioral state change in response to internal and external stimuli experienced along its path". Line 60: "characterize the velocity and the tortuosity along a movement path". Here, do you infer that "an individual’s behavioral state changed in response to internal and external stimuli experienced along its path" by means of "characterizing the velocity and the tortuosity along a movement path"? I believe you should consider stating that clearly in a sentence here.
 Lines 368—385: Could you please consider calling the supplementary tables and figures in this whole paragraph, for the sake of clarity?
 Line 384: If I understood well, the 'exploratory behavior' is a search process for food as it also is the 'local search' state, and then I believe you could add any comment about the way to better distinguish both states in further studies.
 Line 479: References are not well formatted.
 Figures 3—5 and Table 1: Please verify additional comments made at the end of the manuscript in PDF.
 Supplementary material: Please verify additional comments made at the end of the manuscript in PDF.
 References: There are no references to the following citations: Desbiez et al. 2021a; 2021b; 2021c; 2021d. Also, there are no citations in the text of the following references: “Desbiez ALJ, Massocato Gabriel F., Attias N, Cove M V. 2020” and “Desbiez ALJ, Massocato Gabriel Favero, Kluyber D. 2020”.

---

## Round 0.2 · Minor Revisions

Thank you for carrying out careful revsions to the manuscript. The reviewers are most happy with those changes. However there are still some points that need to be addressed. Reviewer 1 is correct - details of capture should be included. Reviewer 2 has some minor editing suggestions. This text should be in the Acknowledgements, and not in the Methods. "We are grateful to the Fazenda Baía das Pedras owners (R. and C. Jurgielewicz) for their hospitality, generous support, and permission to work on their land." I look forward to seeing your revised manuscript.

·

Basic reporting

The English language is still very good.

I believe the revised title is now clearer and more representative of the actual content of the article.

The references appear to have been thoroughly checked. The citation format is now consistent throughout the manuscript.

Experimental design

I thank the authors for providing more information regarding the anesthesia procedure. However, why not provide information regarding the capture also? Unless a matter of word count, I believe this could be an interesting addition. After reading (and trying to translate following the authors’ advice) the Ministry’s authorization, I understand that the animals were either captured manually or using a “funnel trap”? How were the animals lured?

I thank the authors for providing a link to the tracking data. I have now been able to access it.

The authors have answered my question regarding the possibility to include a picture of an animal wearing the device. I agree with their explanation.

Validity of the findings

If I follow the lines’ number provided in the PDF version of the manuscript (peerj-reviewing-75326-v1), it appears to me that the lines do not match with the ones provided in the Author checklist document. For example, Ethical statement is reported to be lines 165-168 when in the manuscript I read it lines 146-149. I advise the authors to verify the checklist to make sure it matches the revised version of the manuscript.

The figures are relevant and informative. The changes made regarding which figures/tables should be in the main manuscript or the supplemental material are explained by the authors.

Additional comments

The revised version of this article appears to be responsive to the previous critique.
My previous comments regarding the interest and importance of this study still stand.

Reviewer 2 ·

Basic reporting

The authors have done all the modifications to the format of the manuscript, including changes in the references style in order to adjust to the journal guidelines.

Experimental design

The methods have been clarified and mre detail has been added in what regards to the management of the armadillos as suggested.

Validity of the findings

The study is novel and has some meaningful results that can be used as a basis for future studies.
The conclusions do not seem to answer the three aims proposed by the authors at the end of the introduction section, I would suggest to review this section in accordance. Also the first part of the conclusions is more a statement of the problematic than a conclusion.

Additional comments

I think the authors have done a great job going through all the suggestions provided. I have two additional minor comments:
Line 148 please use anesthetized instead of sedated
Line 148 please include "an internal" before "VHF radio...."
Line 165 include a , after Kluyber et al. 2020

·

Basic reporting

no comment

Experimental design

no comment

Validity of the findings

no comment

Additional comments

I would like to thank all the authors for accepting some of my suggestions. By having done so, I believe that the manuscript has improved a lot. It is a beautiful piece of work. Congratulations.

---

## Round 0.3 · accepted · Accept

Thank you for the final revisions.